# Energy-Efficient Decentralized Broadcasting in Wireless Multi-Hop Networks [note 1]

**DOI:** 10.3390/s23177419

**Published:** 2023-08-25

**Authors:** Artur Sterz, Robin Klose, Markus Sommer, Jonas Höchst, Jakob Link, Bernd Simon, Anja Klein, Matthias Hollick, Bernd Freisleben

**Affiliations:** 1Department of Mathematics & Computer Science, Philipps-Universität Marburg, 35043 Marburg, Germany; msommer@informatik.uni-marburg.de (M.S.); hoechst@informatik.uni-marburg.de (J.H.); freisleb@informatik.uni-marburg.de (B.F.); 2Department of Computer Science, Technical University of Darmstadt, 64289 Darmstadt, Germany; rklose@seemoo.de (R.K.); jlink@seemoo.de (J.L.); mhollick@seemoo.de (M.H.); 3Department of Electrical Engineering & Information Technology, Technical University of Darmstadt, 64283 Darmstadt, Germany; b.simon@nt.tu-darmstadt.de (B.S.); a.klein@nt.tu-darmstadt.de (A.K.)

**Keywords:** wireless networks, data dissemination, broadcast tree

## Abstract

Several areas of wireless networking, such as wireless sensor networks or the Internet of Things, require application data to be distributed to multiple receivers in an area beyond the transmission range of a single node. This can be achieved by using the wireless medium’s broadcast property when retransmitting data. Due to the energy constraints of typical wireless devices, a broadcasting scheme that consumes as little energy as possible is highly desirable. In this article, we present a novel multi-hop data dissemination protocol called BTP. It uses a game-theoretical model to construct a spanning tree in a decentralized manner to minimize the total energy consumption of a network by minimizing the transmission power of each node. Although BTP is based on a game-theoretical model, it neither requires information exchange between distant nodes nor time synchronization during its operation, and it inhibits graph cycles effectively. The protocol is evaluated in Matlab and NS-3 simulations and through real-world implementation on a testbed of 75 Raspberry Pis. The evaluation conducted shows that our proposed protocol can achieve a total energy reduction of up to 90% compared to a simple broadcast protocol in real-world experiments.

## 1. Introduction

In several areas of wireless networking, such as wireless sensor networks (WSN) or the Internet of Things (IoT), application data need to be disseminated across multiple devices to reach their destination. In particular, there are applications in which one device needs to disseminate data to all other nodes in the network, e.g., in tasks such as network configuration [1], update dissemination [2], or network diagnosis. Other typical examples are announcements, notifications, and event distribution [3].

To enable these applications, the wireless medium’s broadcast property can be utilized. This means that multiple devices in the vicinity of the sender can receive a transmitted packet. Since the possible transmission range of a wireless device is limited by its maximum transmission power, a multi-hop data dissemination scheme is required, i.e., nodes in different locations retransmit the received packets to distribute the data over an extended area. Furthermore, since wireless network devices usually have a limited energy budget, only selected nodes should actively participate in data dissemination through retransmission, while the other nodes should be passive consumers of the received packets. Thus, the active nodes form a spanning tree that connects all nodes of the network to the root node. There are several combinations of nodes that can form a spanning tree, but the construction of an energy-minimal spanning tree is NP-hard [4]. Furthermore, to minimize the total energy consumption of a network in a real setting, the practical applicability of a spanning tree construction algorithm is important. However, although there are several spanning tree approaches in the literature, virtually none of them can be implemented in a real-world environment due to different shortcomings. Many approaches assume global knowledge of parameters or states of nodes, but such knowledge is not available without additional effort in practice. Most works do not consider the potential occurrence of graph cycles. Some studies assume a pre-existing spanning tree and omit the initial spanning tree construction phase. Others do not optimize the transmission power levels of the individual nodes, resulting in missed potential. Centralized approaches are generally not suitable for wireless ad hoc and multi-hop networks.

In this article, we present the Broadcast Tree Protocol (BTP), a novel multi-hop data dissemination protocol that constructs a spanning tree in a decentralized manner such that the total energy consumption in a network is minimized. This is achieved by letting each node connect to the broadcast tree in a way that minimizes its own contribution to the total energy consumption. To realize this approach, we first model the broadcast tree using a game-theoretical model (based on the work conducted by Mousavi et al. [5]) that has been proven to converge to a Nash Equilibrium, yielding better results than other approaches from the literature. However, the original model makes assumptions that render a practical implementation impossible. First, it assumes that every node is aware of the transmission power of its neighbors at all times. However, this assumption translates to complicated and expensive status updates between neighboring nodes. Second, the original model assumes that all nodes make their decisions at the same time. Achieving this in practice would require the precise time synchronization of all nodes, which, in itself, is a challenging task. Third, the potential game proposed by Mousavi et al. [5] uses a weakly dominant strategy where a node can switch between different parent nodes even if the energy costs that it incurs stay the same. This strategy can result in a ping-pong effect, i.e., a node may permanently switch between multiple parent nodes, while the overall transmission power of the network is not reduced further. Fourth, the original model assumes that the constructed broadcast tree does not have cycles. However, due to the lack of time synchronization of all nodes and the lack of global knowledge at each node, graph cycles cannot be easily avoided in practice. Therefore, we present an approximation of the original model that retains the convergence properties of the original model and can be implemented in real-world settings.

In particular, we make the following contributions:We present BTP, a practical protocol that approximates a game-theoretical model for constructing an energy-minimal broadcast tree while preserving both its convergence and Nash Equilibrium properties. To the best of our knowledge, BTP is the first protocol for energy-efficient data dissemination in wireless multi-hop networks based on a provably optimal game-theoretical model that is implemented on real hardware.We design and implement a discovery mechanism that allows the nodes in a wireless multi-hop network to construct a broadcast tree in a decentralized fashion using locally available information from their direct neighbors.We change the decision strategy of the algorithm from a *weakly* dominant strategy to a *strictly* dominant strategy to avoid ping-pong effects, in which a node may potentially continue changing its decision without further reducing the transmission power.We implement and evaluate three different algorithms for inhibiting graph cycles. Specifically, (1) the Path-to-Source algorithm avoids cycles by letting each node keep track of the path from the root to itself so that each node can check the consistency of the spanning tree when making decisions. (2) The Mutex algorithm avoids cycles by letting each node lock its sub-tree when connecting to a different parent node, ensuring consistency at all times. (3) The Ping-to-Source algorithm allows for cycles temporarily, but it detects and resolves such cycles immediately.We evaluate BTP using different tools to assess its feasibility under various conditions. First, we use Matlab simulations to compare BTP against approaches from the literature. Second, we perform NS-3 simulations to investigate the scalability of BTP. Third, we present a real-world implementation of BTP, that is evaluated on a testbed of 75 Raspberry Pis deployed in one of our university buildings to explore its practical feasibility. The evaluations show that BTP can achieve an energy reduction of up to 90% in real-world experiments compared to a simple broadcast protocol.The code of the NS-3 implementation and the real-world implementation has been released under a permissive open-source license. Furthermore, all code required to reproduce the experiments as well as the experimental artifacts are also been made available.

This article is organized as follows. Section 2 discusses related work. Section 3 introduces our system model. Section 4 presents the game-theoretical model, followed by design of the formal protocol. Section 5 presents comparisons of BTP against other algorithms in Matlab simulations, while Section 6 explores the scalability of BTP in NS-3 simulations. Section 7 presents the implementation of BTP on Raspberry Pis and an evaluation through practical experiments. Section 8 concludes this article and outlines areas for future work.

## 2. Related Work

Several approaches have been proposed for designing energy-efficient broadcast trees. In one approach, the nodes broadcasted beacon packets with increasing transmission power to iteratively build a set of all their neighbors [6]. In another work, each node was connected to the node for which the minimal additional energy usage was required [7]. In yet another work, each node computed its optimal parent under a certain optimization objective, while the parent was on the path to the source node [8]. Energy accumulation to minimize the nodes’ transmission power was also used [9], while others attempted to find a spanning tree that minimizes the number of transmissions and the transmission delay at the same time [10]. Also, clusters of trees have been built using different methodologies [11,12,13]. Several approaches use game theory to construct a spanning tree that requires minimal transmission power [14,15]. Finally, some authors have attempted to minimize the path lengths from the source to all other nodes to minimize energy consumption [16,17].

Furthermore, several centralized approaches have been proposed. For example, there are approaches in which sensor nodes send a beacon packet to a central controller that constructs the tree [18] or minimizes the path lengths of the tree [19].

Several approaches do not rely on a tree structure to disseminate data in wireless ad hoc networks but still attempt to minimize total energy consumption by, for example, proposing an energy-efficient sensor placement algorithm [20] or by letting a controller decide which nodes should transmit the data [21,22]. Other approaches involve the formulation of a linear programming problem that finds the most energy-efficient unicast path to reach all nodes [23], the proposal of a data-forwarding scheme that utilizes nodes’ contextual information [24], or the use of a k-coverage algorithm to distribute data energy-efficiently in wireless underwater sensor networks [25].

Additionally, some approaches do not minimize energy consumption but optimize throughput [26], delivery probability [27], end-to-end latency [28,29,30], or fairness [31].

Furthermore, practical approaches with an implementation were either aiming not to minimize energy consumption but other metrics such as path length or latency [32,33] or were not based on a proven theoretical model [34,35,36].

Finally, apart from approaches that rely on topology control or spatial placement, energy efficiency can also be optimized through techniques like satellite communication [37,38], beamforming [39,40,41], or MU-MIMO [41]. However, these approaches are beyond the scope of our work since devices used in multi-hop wireless sensor networks are typically not equipped with hardware capable of using such techniques.

Compared to BTP, the aforementioned decentralized approaches suffer from various problems. Many approaches require global knowledge of parameters or the states of neighboring nodes and do not explicitly address the inhibition of graph cycles. Some works do not consider the initial tree construction phase but assume that an already-constructed tree exists, where, afterwards, only the transmission power is adjusted. Other approaches also squander their potential by not leveraging the possibility of adjusting the transmission power of each node. Furthermore, centralized approaches are usually not suitable in the area of wireless ad hoc and multi-hop networks. Spanning trees offer the possibility to reach all nodes in a network with a minimum amount of energy, making approaches that do not rely on a tree structure questionable in terms of minimizing the energy consumption. Finally, the presented approaches either do not provide an implementation or a real-world evaluation using off-the-shelf Wi-Fi devices or are not based on a proven theoretical model. In fact, most of the presented works only propose a theoretical model without considering its applicability. Some of these models cannot be implemented under real-world constraints.

## 3. System Model

Our system model is based on the assumption that a given source node contains data to send to all other nodes in a network. The nodes are spatially distributed, and each node has a single antenna and a maximum transmission power of $p^{\max}$. Due to the path loss properties of wireless transmissions, the source node may not be able to reach all other nodes, even if transmitting with $p^{\max}$. Therefore, a multi-hop transmission scheme is required. Table 1 lists the mathematical notations of our system model described below.

### 3.1. Graph Representation

A rooted spanning tree is an appropriate graph for a multi-hop broadcast scheme. In the remainder of this article, we call such a tree a *broadcast tree*. A broadcast tree is defined as a graph T=(V,E), where the vertices *V* correspond to all nodes of the network, the source node S∈V is the root of the broadcast tree, and the edges *E* are the connections between the nodes. Each edge e∈E has a weight $p_{i,j}$ corresponding to the transmission power required to successfully establish a wireless communication link between the two nodes $\left\{i,j\right\}\in V_{e}$ connected via *e*. Furthermore, a broadcast tree must not contain any cycles, i.e., for a given broadcast tree T=(V,E), there is no path of edges $\left(e_{1},e_{2},\ldots ,e_{n}\right)$ with a vertex sequence $\left(v_{1},v_{2},\ldots ,v_{n},v_{1}\right)$.

In a broadcast tree, each node j∈V∖{S} has exactly one parent *i*, whereas each parent may have multiple children. The set of children of a parent *i* is denoted as $C_{i}$, as indicated by the blue box in Figure 1. Further, we assume that all communication links are bi-directional. Due to the broadcast property of wireless communications, a parent node *i* has to send data only once, while all its children $C_{i}$ should be able to receive it.

### 3.2. Transmission Power Model

To establish a connection between a parent *i* and all its children $C_{i}$, node *i* has to transmit data with enough transmission power to ensure that the received signal strength exceeds a certain level at all of its children $C_{i}$. The signal-to-noise-ratio (SNR) $\gamma_{j}$ at receiver $j\in C_{i}$ is:(1)$$\gamma_{j}=\frac{p_{i}{\left|h_{i,j}\right|}^{2}}{\sigma^{2}}$$Here, $p_{i}$ is the transmission power that parent *i* uses to send the data, $|h_{i,j}{|}^{2}$ is the channel gain, and $\sigma^{2}$ is the noise power. Furthermore, based on the minimally required SNR $\gamma^{min}$, the minimum transmission power that *i* must use to reach *j* is as follows:(2)$$p_{i,j}=\frac{\gamma^{min}\sigma^{2}}{|h_{i,j}{|}^{2}}$$In Figure 1, this is visualized as $p_{S,i}$ on the connection between *S* and *i*. Furthermore, a node *j* that receives a signal from node *i* can calculate the required transmission power as follows:(3)$$p_{i,j}=\frac{p_{i}\gamma^{min}}{\gamma_{j}}$$

The transmission power that a parent node *i* must use to reach all its children $C_{i}$ depends on its most distant child and is generally bound by $p^{\max}$:(4)$$p_{i}\left(C_{i}\right)=\max_{j\in C_{i}}\left(p_{i,j}\right)\leq p^{\max}$$Note that both connections of *S* are marked with $p_{S}\left(C_{S}\right)$ in Figure 1 due to the broadcast property. The neighborhood of a parent *k* is defined in the following manner:(5)$$N_{k}=\left\{l|l\in V,p_{k,l}\leq p^{\max}\right\},$$
i.e., it contains all nodes that can be reached by parent *k* with the transmission power $p^{\max}$ or less, as represented by the green box in Figure 1.

Our goal is to minimize the total transmission power in the network.
(6)$$p=\sum_{i\in V}p_{i}\left(C_{i}\right)$$The total transmission power is defined as the sum of the transmission powers $p_{i}\left(C_{i}\right)$ of all nodes i∈V for transmitting to their respective child nodes $C_{i}$.

## 4. Broadcast Tree Protocol

This section presents the design of our Broadcast Tree Protocol (BTP) that is based on a game-theoretical model.

### 4.1. Potential Game

Our approach is based on a *potential game* [42], i.e., all nodes cooperate to minimize the total transmission power (Equation (6)) required to disseminate data over an entire broadcast tree *T*. The use of potential games to calculate energy-efficient broadcast trees in a decentralized manner has been proposed by Mousavi et al. [5].

#### 4.1.1. Design of the Potential Game

Our potential game is designed as a *child-driven* game, meaning that the receiving nodes (children) decide which transmitting node they select (parents). The construction of the broadcast tree *T* is executed in iterations, and the current iteration number is denoted by the index *t*. The potential game G is described by a set of rational players P containing all destination nodes j∈V∖{S}, a set $A_{j}$ of possible actions for each player *j*, and a player-specific local cost function $\varphi_{j}$ for each player *j*.

Each node *j* individually decides from which parent node it should receive data. Therefore, the set $A_{j}^{\left(t\right)}$ of possible actions for each player *j* is the set $A_{j}^{\left(t\right)}=V\setminus \left\{j\right\}$ with its potential parents. In each iteration *t*, each node j∈P selects a parent node, which is denoted by the action $a_{j}^{\left(t\right)}\in A_{j}^{\left(t\right)}$. The action profile of the game $a^{\left(t\right)}=\left(a_{1}^{\left(t\right)},\cdots ,a_{n}^{\left(t\right)}\right)$ is a vector containing the actions of all nodes in iteration *t*, and $a_{-j}^{\left(t\right)}$ represents the actions of all nodes except the *j*-th node. Each node *j* has a player-specific local cost function $\varphi_{j}\left(a_{j}^{\left(t\right)},a_{-j}^{\left(t\right)}\right)$, which depends on the node’s action $a_{j}^{\left(t\right)}$ and the actions $a_{-j}^{\left(t\right)}$ of all other nodes in the network.

This cost function, $\varphi_{j}$, is designed to be the marginal contribution of the transmission power required by its parent *i* to reach the considered node *j*:(7)$$\varphi_{j}\left(i,a_{-j}^{*}\right)=p_{i}\left(C_{i}\right)-p_{i}\left(C_{i}\setminus j\right)$$

To describe the solutions of G, we introduce the concept of a Nash Equilibrium (NE). An action profile $a^{*}$ is an NE of G if
(8)$$\varphi_{j}\left(a_{j}^{*},a_{-j}^{*}\right)\leq \varphi_{j}\left(a_{j},a_{-j}^{*}\right),\;\forall j\in P,a_{j}\in A_{j}^{\left(t\right)}$$
holds, i.e., no node in P can reduce its local cost $\varphi_{j}$ any further by changing its action $a_{j}$. We consider the best response of node *j* in iteration *t* to the other nodes’ actions as follows:(9)$$a_{j}^{\left(t\right)}=\underset{a_{j}^{\left(t\right)}\in A_{j}^{\left(t\right)}}{\mathrm{argmin}}\;\;\;\varphi_{j}\left(a_{j}^{\left(t\right)},a_{-j}^{\left(t\right)}\right).$$

Mousavi et al. [5] proved that if all players select their actions according to their best response (Equation (9)), the strategy profile will converge to an NE, which minimizes the transmission power over all nodes (Equation (6)).

Although the presented potential game contributes to the decentralized construction of energy-efficient broadcast trees, its practical implementation poses several challenges. First, in the potential game, it is assumed that every node *j* knows all potential parents in *V* at any given iteration *t* as well as the actions of all other players k∈P. However, this information is not easily available in a real-world system; therefore, it must be exchanged and maintained. Second, the potential game assumes iterations with discrete time steps, while every node makes a decision at each time step. This is not feasible for a distributed algorithm in the real world since there is no mechanism to precisely synchronize all nodes. Therefore, each node must make a decision whenever it obtains information asynchronously from its neighborhood. Third, the potential game follows a weakly dominant strategy, where a child node *j* changes parents if the total transmission power *p* remains, at most, the same. An alternative is a strictly dominant strategy, where a child node *j* only switches to another parent node if the total transmission power *p* is strictly reduced. In a real-world implementation, a problem may arise wherein no stable broadcast tree *T* is found when a child node can reach two potential parent nodes with the same transmission power, causing the child node to repeatedly switch between the two parent nodes. Fourth, while the potential game assumes that there are no graph cycles, a real-world implementation needs a mechanism to ensure this property. Thus, we modify the potential game as follows.

#### 4.1.2. Approximation of the Potential Game

In contrast to the original approach, (t) now denotes the iteration of a node *j* that is not synchronized with other nodes. Therefore, in any given iteration (t), a node *j* possesses the information about the transmission power $p_{Q_{j},j}$ that *j*’s current parent $Q_{j}$ needs to reach node *j*. Furthermore, for a given potential parent *i* at iteration (t), *j* knows the transmission power required by *i* to reach all of its children $C_{i}$, i.e., $p_{i}\left(C_{i}\right)$, since node *i* shares this information through broadcast messages. Therefore, in iteration (t), *j* has two possible actions ${A^{\prime }}_{j}^{\left(t\right)}=\left\{Q_{j},i\right\}$: either stay with the current parent $Q_{j}$ or switch to *i*. Furthermore, *j* can also derive the transmission power $p_{i,j}$ that *i* needs to reach *j*, as shown in Equation (3). With the information received from $Q_{j}$ and *i*, *j* can also derive how much $Q_{j}$ can potentially reduce its transmission power if *j* is no longer a child of $Q_{j}$ as well as how much *i* would need to increase its transmission power to reach all its children if *j* was a child of *i*. With this information in hand, *j* has two cost functions that express its marginal contribution to *i* and $Q_{j}$, respectively: (10)$$\begin{array}{cc} \varphi_{j}^{\left(t\right)}\left(i\right) & =p_{i}\left(C_{i}\cup j\right)-p_{i}\left(C_{i}\right) \end{array}$$(11)$$\begin{array}{cc} \varphi_{j}^{\left(t\right)}\left(Q_{j}\right) & =p_{Q_{j}}\left(C_{Q_{j}}\right)-p_{Q_{j}}\left(C_{Q_{j}}\setminus j\right) \end{array}$$

Using this information, and changing the weakly dominant strategy to a strictly dominant strategy, a node *j* switches to the potential parent *i* if the marginal contribution to *i* is lower than the marginal contribution to the current parent $Q_{j}$: (12)$$a_{j}^{\left(t\right)}=\underset{a_{j}^{\left(t\right)}\in {A^{\prime }}_{j}^{\left(t\right)}}{\mathrm{argmin}}\;\;\;\varphi_{j}\left(a_{j}^{\left(t\right)}\right)$$

This modification enables the child node to make decisions whenever a new potential parent node is discovered instead of having to exchange information with all other nodes. To solve the problem with our approximation in which a child node *j* does not know all the potential parents in *V*, each node needs more than one turn in the game to find the optimal parent. Therefore, our last modification is the introduction of a counter that tracks how often a potential parent has been discovered without *j* switching parents. Once a threshold is reached, we consider all parents of a node to be discovered and the node to be finished. With this last change, child *j* might not find the optimal parent in the first iteration, but, over time, it will switch parents until the best parent is found.

In the following, we show that the modified game still converges to an NE in finite time. We make the following assumptions: (1) nodes use discovery packets to identify potential parents, no discovery packets are lost, and all nodes receive discovery packets from their entire neighborhood $N_{k}$; (2) the obtained values $p_{i}\left(C_{i}\cup j\right),p_{i}\left(C_{i}\right),p_{Q_{j}}\left(C_{Q_{j}}\right)$, and $p_{Q_{j}}\left(C_{Q_{j}}\setminus j\right)$ are undisturbed. The first assumption requires the counter mentioned above to be set to a reasonably high value, which is carried out empirically in the evaluation in Section 7.3. The second assumption might seem impractical, but our evaluation shows that noise only affects the obtained values marginally and does not cause frequent decision changes.

Mousavi et al. [5] have shown that an NE exists for the original game. We show that BTP converges to an NE of the modified game. For this purpose, we use the concept of an improvement path [43], which is defined as a sequence of action profiles $\left\{a^{\left(0\right)},a^{\left(1\right)},\cdots \right\}$, where in $a^{\left(t+1\right)}$, every node *j* that changes its action $a_{j}^{\left(t+1\right)}\neq a_{j}^{\left(t\right)}$ has a lower cost function $\varphi_{j}\left(a_{j}^{\left(t+1\right)}\right)<\varphi_{j}\left(a_{j}^{\left(t\right)}\right)$. Furthermore, at least one node needs to choose the same action in $a^{\left(t\right)}$ and $a^{\left(t+1\right)}$. Using BTP, the action profile $a_{j}^{\left(t\right)}$ of the nodes follows an improvement path. This can be directly seen by referring to the best response strategy (Equation 12), in which nodes only change their actions if their cost function decreases. For potential games, the finite improvement path property holds, leading to an NE in a finite amount of time [43]. Therefore, BTP converges to an NE in a finite number of steps.

### 4.2. BTP

BTP is based on the game-theoretical model presented in Section 4.1. BTP consists of two phases: (1) the broadcast tree construction phase, where the source node *S*, i.e., the node intending to disseminate data to all other nodes, initiates the construction of the broadcast tree *T* (this phase ends with a tree topology, where the source node *S* is the root), and (2) the data dissemination phase, where the actual data are sent from *S* to all other nodes.

The broadcast tree *T* consists of the source node *S*, which is the root of the broadcast tree, and parent and child nodes. Here, all parents except *S* are also children, and all children except the leaves are also parents. Since this approach is child-driven, parents only broadcast their own state information periodically, and children use this information to decide which node they will choose to be their parent. Furthermore, a child may choose a node as its parent, but a chosen parent node may or may not accept parenthood for the given child. In the latter case, the child must find another parent.

#### 4.2.1. Broadcast Tree Construction Phase

To initialize the decentralized broadcast tree construction phase, the source node *S* sends a Neighbor Discovery packet using the maximum transmission power $p^{\max}$. Every node i∈V that is already part of the broadcast tree *T* also periodically broadcasts Neighbor Discovery packets with maximum transmission power $p^{\max}$. Every receiver $j\in N_{i}$ of a Neighbor Discovery packet checks whether the sender *i* of the received packet is a suitable parent *Q*. Two cases may occur. In the first case, *j* is not connected to any parent, e.g., during the initial construction of the broadcast tree. In this case, *j* requests *i* to become *i*’s child by sending it a Child Request packet, which it, in turn, may or may not accept, as described later. Second, *j* is already a child and is connected to parent $Q_{j}$, which is different from *i*. In this case, *j* checks if switching from $Q_{j}$ to *i* would decrease *p* (see Equation (6)), as shown in Section 4.1.2. To this end, *j* must check whether switching to having node *i* as its parent would reduce the transmission power of $Q_{j}$ to a greater extent than *i* would have to increase its transmission power to reach all its children. To do so, *j* needs four transmission power values (see Section 4.1.2):$p_{Q_{j}}\left(C_{Q_{j}}\right)$, i.e., the transmission power of $Q_{j}$ needed to reach all its children;$p_{Q_{j}}\left(C_{Q_{j}}\setminus \left\{j\right\}\right)$, i.e., the transmission power of $Q_{j}$ if *j* is no longer $Q_{j}$’s child;$p_{i}\left(C_{i}\right)$, i.e., the transmission power of *i* needed to reach all its current children;$p_{i}\left(C_{i}\cup j\right)$, i.e., the transmission power of *i* if *j* becomes *i*’s child.

To provide these values to all potential children of a node *i*, every BTP packet includes the transmission power $p_{i}\left(C_{i}\right)$, as well as the transmission power that would be required to reach the second-farthest child of *i*, along with its respective address. When node *j* receives a BTP packet from node *i*, it can additionally calculate the transmission power $p_{i,j}$ that *i* would need to reach *j* by means of Equation (3). Thus, *j* can also calculate $p_{i}\left(C_{i}\cup j\right)$. When *j* knows the four values, it switches to *i* if the following condition holds:(13)$$p_{i}\left(C_{i}\cup j\right)-p_{i}\left(C_{i}\right)<p_{Q_{j}}\left(C_{Q_{j}}\right)-p_{Q_{j}}\left(C_{Q_{j}}\setminus \left\{j\right\}\right)$$
i.e., if the current parent $Q_{j}$ can reduce its transmission power to a greater extent than the new parent *i* must increase its own transmission power.

When node *j* decides to choose node *i* as its new parent, it sends a Child Request packet to *i* in order to become its child. Node *i* verifies that *j* is not *i*’s parent, *j* is not already a child of *i*, and *j* is reachable from *i*, i.e., $p_{i,j}\leq p^{\max}$. If all these checks are successful, *i* accepts *j* as a child using a Child Confirmation packet or rejects it otherwise using a Child Rejection packet. When *i* accepts *j*, it adjusts its transmission power to $p_{i}\left(C_{i}\cup j\right)$, resulting in an increase if $p_{i,j}$ is greater than $p_{i}\left(C_{i}\right)$; otherwise, the value stays the same as before. Even if *i* must increase its transmission power, the total transmission power *p* is still lower because *j* only connects to *i* if the condition of Equation (13) holds. When *i* accepts *j* as its child, *j* disconnects from its old parent $Q_{j}$ using a Child Revocation packet. $Q_{j}$, in turn, removes *j* from its child list $C_{Q_{j}}$ and adjusts its transmission power accordingly to reach the farthest child $k\in C_{Q_{j}}\setminus \left\{j\right\}$. However, when *i* rejects *j*, *j* places *i* on a blocklist to avoid repetitive connection attempts and repeats the above processes to find a new parent.

Finally, since *S* does not know the global state of the network, it cannot know when the broadcast tree has reached its optimal state. Therefore, each node maintains a counter that tracks the iterations without any changes, i.e., without connecting to or disconnecting from parents or adding or removing children. As soon as this counter reaches a threshold, *j* considers itself finished and notifies its parent $Q_{j}$ using an End of Construction packet. When $Q_{j}$ has received such a packet from all its children $k\in C_{Q_{j}}$ and when it is itself finished, it notifies its own parent that it has finished its game. This procedure continues until *S* has received End of Construction packets from all its children $l\in C_{S}$. The broadcast tree construction phase is then finished, and the data dissemination phase starts. Once the broadcast tree is constructed, each node *i* sets its transmission power to $p_{i}\left(C_{i}\right)$ so that *i* just barely reaches all its children $C_{i}$. Thus, the total transmission power is minimized.

During the broadcast tree construction phase, graph cycles may potentially occur, which must either be avoided or detected and broken up. Graph cycles can occur in three cases. In the first case, the source node *S* may try to connect to another node as its parent. Since *S* is defined as the root of the broadcast tree, this eventuality must be avoided, which can easily be accomplished. In the second case, a parent node *i* may try to connect to one of its children $j\in C_{i}$. To circumvent this eventuality, *i* must check if $j\in C_{i}$ and refrain from connecting to children. In the third case, a parent node *i* may try to connect to a node *k* that is not its direct child but that includes *i* on the path from *k* to *S*$\left(S,\ldots ,i,\ldots ,Q_{k},k\right)$. To handle this case, we propose two cycle avoidance algorithms and one cycle detection algorithm.

Path-to-Source

In the Path-to-Source cycle avoidance algorithm, every node *j* that successfully connects to a parent $Q_{j}$ adds its own address to a list of addresses $\left(S,\ldots ,Q_{j},j\right)$ that represents the entire path from source *S* to node *j*. This list is included in each Neighbour Discovery packet. When a node *k* tries to connect to *j*, it first has to check if $k\in (S,\ldots ,Q_{j},j)$. If it is part of the path, the node must not try to connect to *j* in order to avoid a cycle.

Mutex

The Mutex cycle avoidance algorithm essentially ensures that the entire broadcast tree is in a consistent state at any point in time. To this end, a node *j* that decides to connect to a node *i* first locks its own sub-tree by notifying all its children $C_{j}$. In this process, all nodes in $C_{j}$ also lock their respective sub-trees until all nodes below *j* are locked. The nodes that are included in a locked sub-tree are not allowed to change their parents or to accept new children until *j* unlocks them again. When *j* tries to connect to *i* and *i* is locked by *j*, *j* detects this situation and refrains from connecting to *i* to avoid giving rise to a cycle.

Ping-to-Source

The Ping-to-Source algorithm is a cycle detection algorithm. After connecting to parent *i*, node *j* sends a unicast Ping-to-Source packet to *i*, which, in turn, forwards the packet to its own parent $Q_{i}$. This process, in which nodes forward the Ping-to-Source packet to their parents, continues until one of three cases occurs. In the first case, the source node *S* receives the message. This occurs when there is no cycle, so *S* can drop the Ping-to-Source packet. In the second case, the Ping-to-Source packet arrives at an intermediate node that has no parent, in which case the packet is also dropped. In the third case, the Ping-to-Source packet eventually arrives at node *j*, which occurs if the broadcast tree has a cycle. In this case, *j* disconnects from *i*. Furthermore, if *j* was connected to a parent $Q_{j}$ before trying to connect to *i*, it would attempt to reconnect to $Q_{j}$, which triggers the entire connection process discussed above.

#### 4.2.2. Data Dissemination Phase

During the data dissemination phase, the source node *S* starts sending the data to its children $l\in C_{S}$ with a transmission power of $p_{S}\left(C_{S}\right)$ (see Equation (4)) using Application Data packets. All nodes *l*, in turn, relay the data to their respective children with a transmission power of $p_{l}\left(C_{l}\right)$. The Application Data packets contain a sequence number. Since the data size may exceed the size of a frame, e.g., 2304 bytes for Wi-Fi, the source node splits the data into chunks and increases the sequence number for every frame accordingly. Since node *i* transmits with a power of $p_{i}\left(C_{i}\right)$, other nodes $j\notin C_{i}$ may still receive the data because they may be reached by *i* when $p_{i,j}\leq p_{i}\left(C_{i}\right)$. In this case, *j* can still process and utilize such receptions as the packets are unambiguous due to their sequence number.

### 4.3. Protocol Packets

BTP uses several packet types for different purposes. Each BTP packet is sent with a transmission power of $p^{\max}$ during the broadcast tree construction phase, while Application Data packets are sent with a transmission power of $p_{i}\left(C_{i}\right)$ by each node i∈V during the data dissemination phase. The following packet types exist in BTP:

Neighbor Discovery

Each node periodically broadcasts Neighbor Discovery packets to inform its neighbors of its presence. When a node receives a Neighbor Discovery packet, it checks whether the sender is a possible new parent node, as presented in Section 4.2.1. In particular, each Neighbor Discovery packet contains information about the transmission power levels required by its sender, as described in Section 4.2.1, allowing each receiver to decide whether to switch parents according to Equation (13).

Child Request

Once a node has identified a possible parent node, it requests a connection by sending a Child Request packet as a unicast message to that parent node.

Child Confirmation

A potential parent node that is able to accept a child request from another node sends a Child Confirmation packet in response as a unicast packet. When the child node receives the Child Confirmation packet, the child node and the parent node are considered to be connected.

Child Rejection

A potential parent that cannot accept a child request from another node sends a Child Rejection packet. A requesting child node that receives such a packet blocks this potential parent and tries to connect to another potential parent.

Child Revocation

Child nodes must inform their parent nodes when they want to disconnect, which is executed by sending a Child Revocation packet as a unicast message to the parent node. This packet type is used after successfully switching to a new parent or when detecting a cycle.

End of Construction

When a node ends the broadcast tree construction phase, it notifies its parent node by sending an End of Construction packet. This information is important for the parent node because the parent must wait for all its child nodes before it is allowed to finish the broadcast tree construction phase itself. As soon as a node finishes the broadcast tree construction phase, it sets a corresponding flag *F* in all its sent packets.

Application Data

In the data dissemination phase, application data are transmitted using the Application Data packet type. A node i∈V sends all data packets with $p_{i}\left(C_{i}\right)$, i.e., with the transmission power that is required to reach its most distant child.

## 5. Matlab Simulation

This section presents Matlab simulations of BTP and other algorithms from the literature, allowing us to compare BTP with other algorithms. However, while simulations allow for control over various parameters, e.g., distances between nodes, transmission power, and channel gain, they do not always reflect the real world accurately.

### 5.1. Experimental Setup for the Matlab Simulation

Table 2 shows the parameters of our Matlab simulation. The nodes are randomly placed in a 500 m × 500 m square area, which allows us to assess BTP under various conditions while not taking advantage of an optimized node placement strategy in order to reflect practical constraints. Still, we require each node to have at least one neighbor according to Equation (5), while the maximum transmission power of a node is set to $p^{\max}=20$ dBm. Further, the number of nodes varies between 10 and 90. The source node is chosen randomly for each simulation run. The channel is based on a path-loss model for which $|h_{i,j}{|}^{2}=\frac{1}{d^{\alpha }}$, where *d* is the distance between the nodes *i* and *j* and α is the attenuation exponent, which is set to α=3. The SNR must exceed $\gamma^{min}=$ 10 dB for correct reception, while the noise power $\sigma^{2}$ is set to −90 dBm. The finishing threshold is set to 10 unchanged iterations (see Section 4.2.1) since the broadcast tree did not improve further with a larger threshold in trial experiments. For each parameter combination, 1000 simulation runs were executed, i.e., a total of 63,000 simulations runs. We implemented the following algorithms for comparison with BTP:

Dijkstra

Dijkstra’s algorithm [44] maintains a set of nodes whose shortest distance from the source is known and gradually expands this set until all nodes are included. It iteratively selects the node with the shortest distance, updates the distances to its neighbors, and continues this process until the shortest paths to all nodes have been established. It is important to note that Dijkstra’s algorithm is used in our comparison to construct a spanning tree rather than to route packets to individual nodes.

BIP

Broadcast Incremental Power (BIP) [7] is an iterative algorithm that exploits the broadcast characteristics of wireless channels through centralized control. Starting from the source node, each iteration establishes a connection between the source and another node in the network either by using a direct single-hop connection or a multi-hop connection, thereby extending the sub-tree.

BIPSW

Broadcast Incremental Power with Sweep (BIPSW) [7] is a variation of the BIP algorithm. The efficiency of BIP is enhanced using a so-called sweep operation, which eliminates redundant transmissions in cases where a node can be served by multiple transmitters. This operation transforms the tree constructed using BIP into a spanning tree.

PCP

The Power Control Protocol (PCP) [14] employs an energy rank for each node in the network, which is defined as the energy that a node adds to a given path to the source. Nodes connect to a parent whose path to the source requires the least energy.

BPG

Broadcast trees with Potential Game (BGP) [5] is the original game-theoretical protocol from which BTP is derived. Section 4.1 provides a detailed description of BPG.

SBP

The Simple Broadcast Protocol (SBP) is a variant of BTP that always uses $p^{\max}$ to disseminate data. SBP is also employed for comparisons in the testbed experiments presented in Section 7 and provides an upper bound of the required transmission power.

### 5.2. Results of the Matlab Simulation

Figure 2 and Figure 3 show the total transmission power as a function of the number of nodes in the network as introduced in Equation (6): $p=\sum_{i=1}^{\left|V\right|}p_{i}\left(C_{i}\right)$.

The x-axes show the number of nodes, while the y-axes show the total required transmission power (given in Watts on a logarithmic scale) for all nodes to disseminate data. The different colors denote different algorithms. Both Figure 2 and Figure 3 show the same results, but Figure 3 does not include SBP to better distinguish between the other algorithms. The key indication of the results is that there are four performance categories. First, SBP shows the worst performance, and the more nodes that are added, the worse this performance becomes. While SBP constructs a spanning tree, it always uses the maximum transmission power $p_{max}$, resulting in this poor performance. Second, Dijkstra clearly uses less power than SBP but still more than the other algorithms since it does not consider the broadcast nature of the wireless channel. Instead, Dijkstra builds a spanning tree for unicast connections, resulting in suboptimal node connections in our broadcast scenario. Third, BIP and PCP are better than Dijkstra, but they still perform worse than BTP. On the one hand, BIP still has unnecessary and redundant connections between nodes. On the other hand, PCP builds a fixed action set at startup, which is not further optimized. In contrast, BTP allows each node to update its action in multiple iterations to respond to the actions of other nodes. Fourth, BTP, BPG, and BIPSW show the best performance. The performance of BIPSW is slightly better with fewer nodes, while that of BTP and BPG is slightly better with more nodes. BIPSW essentially outperforms BIP since it removes the redundant connections from BIP’s solution. Furthermore, BIPSW is a centralized algorithm with global knowledge about the links and connections of a network, allowing it to potentially provide better decisions than BTP, which only uses local information. Besides BIPSW, all the algorithms except BTP and SBP also require global knowledge about the entire network. This is not realistic in a real-world scenario.

In summary, the results of the Matlab simulation show that BTP is on par with or better than the other algorithms. BTP is the only algorithm operating under realistic assumptions, using only local information from direct neighbors.

## 6. NS-3 Simulation

In this section, our NS-3 simulation of BTP and the corresponding results are presented. We investigate three aspects. First, while the original potential game has already been compared to alternative approaches in Section 5, in this section, we show that our approximation can minimize the total energy consumption. Second, using the NS-3 simulation, we can evaluate all three cycle avoidance and detection algorithms. Finally, the NS-3 simulation allows us to validate the scalability of BTP.

Although BTP is agnostic with respect to the underlying physical and link layer implementations, we selected Wi-Fi as the underlying wireless technology to allow for the comparison of the results of the NS-3 simulation with a real-world implementation evaluated using real hardware, namely, a Wi-Fi-based testbed (see Section 7). We assume that a data link layer ensures the reliable delivery of packets directly to other nodes. Access to the wireless transmission medium is coordinated through a Carrier Sense Multiple Access (CSMA) mechanism.

NS-3 is an open-source and event-driven simulator with several protocols ready to use, e.g., IP, TCP, or UDP. The individual components of the NS-3 simulator are divided into modules that can be used or extended to set up a self-defined environment.

By exploiting NS-3’s energy framework, it is possible to simulate different energy sources. Based on the datasheets of the MAX28282 (https://datasheets.maximintegrated.com/en/ds/MAX2828-MAX2829.pdf (accessed on 4 July 2023)) and MAX28313 (https://datasheets.maximintegrated.com/en/ds/MAX2831-MAX2832.pdf (accessed on 4 July 2023)) Wi-Fi chips, we approximate the power consumption of a Wi-Fi chip as a function of the transmission power through a polynomial regression:$$\begin{array}{cc} p_{mA}\left(p_{dBm}\right)= & -0.000009708995023p^{5}-0.00089877372p^{4}-0.03112035853p^{3}\\ \; & -0.4798606017p^{2}-2.427503769p+124.4196777 \end{array}$$

This function is essentially used to calculate the energy consumed over a period of time. The energy model of NS-3 tracks state-changes of the Wi-Fi module. If the state changes, for example, from receiving to transmitting, the energy consumed over the duration of the transmitting state is calculated and subtracted from the energy source. Switching losses, additional electronic components, or fluctuations due to heat are not simulated.

In our implementation, we used properties that model IEEE 802.11 characteristics, as follows. In wireless networks, preambles are used to announce the arrival of a packet and allow the receiver to synchronize with a received frame. During preamble detection, the signal-to-noise ratio (SNR) is determined. If the SNR is too low, the packet can be discarded because it may not be possible to decode it without an error. To reproduce the characteristics of Wi-Fi, the noise power of the channel can be modeled as follows:(14)$$\sigma^{2}=k_{B}*290*w+\sigma_{int.}^{2}$$
where $k_{B}$ is the Boltzmann constant, *w* is the channel width in Hz, and $\sigma_{int.}^{2}$ is the noise of interfering transmissions [45]. Furthermore, $\gamma^{min}$, i.e., the minimal SNR that still allows for the decoding of a packet, was set to 4 dB. For path loss $|h_{i,j}{|}^{2}$, we used the YANS model (https://www.nsnam.org/docs/release/3.24/doxygen/classns3_1_1_yans_wifi_channel.html#details (accessed on 4 July 2023)). Further, we set the maximum possible transmission power to $p^{\max}=23$ dBm.

### 6.1. Experimental Setup for the NS-3 Simulation

Table 3 summarizes the parameters used for the evaluation of the NS-3 simulation.

We simulated six different node numbers from 50 to 300 in steps of 50 in five areas of different sizes, ranging from 100 m × 100 m to 500 m × 500 m, which we refer to as area configurations (1) to (5) in the remainder of this article. All nodes were placed randomly within the plane, while a pseudo-random number generator was initialized with a fixed seed and increased with every iteration of an experimental configuration. This allowed us to evaluate the protocols under various conditions, without relying on an optimized node placement strategy. Furthermore, we evaluated all three cycle-handling algorithms presented in Section 4.2.1. The finishing threshold for unchanged game rounds (see Section 4.2.1) was set to 10. Besides BTP, we also evaluated the SBP protocol. A data size of 1 KiB was used, i.e., the payload occupying a single Wi-Fi frame. Finally, each experimental run was executed until the payload was transmitted to all nodes or aborted after 20 s, and each configuration was repeated 30 times. In total, 5400 experimental runs were executed.

### 6.2. Results of the NS-3 Simulation

#### 6.2.1. Total Energy Consumption

We first compare the total energy consumption values of BTP and SBP throughout the entirety of their experiments, i.e., incorporating both the broadcast tree construction phase and the data dissemination phase. The results are shown in Figure 4.

The x-axes denote the simulation area, and the y-axes show the total energy in Joules. The colors denote the different quantities of nodes, while the left and right sub-plots show the results regarding BTP and SBP, respectively. It is evident that BTP requires significantly less energy than SBP. In fact, depending on the experimental configuration, an energy reduction of between 83% and 92% can be achieved when using BTP. Moreover, in the worst case (i.e., 300 nodes, area configuration (1)), BTP requires only about 150 J, and even in the best case (50 nodes, area configuration (1)), with 203 J, it requires less energy, than SBP. This energy reduction primarily results from two effects. First, using a broadcast tree based approach reduced the total number of packets sent in the network. In fact, BTP required about 70% fewer data packets than SBP, resulting in 70% less energy consumption. Second, the remaining 13% to 22% reduction in energy consumption is a result of the optimal broadcast tree. It is also noticeable that the number of nodes is positively correlated with energy consumption, which is reasonable since more nodes need more energy, even if they send with minimal transmission power. The simulation area, however, is negatively correlated with energy consumption, i.e., the larger the area, the lower the total amount of energy consumed. This is due to the fact that an increasing area leads to the greater distances between nodes, thus not becoming part of the broadcast tree, which leads to their inability to disseminate data, thereby reducing overall energy consumption. This means it is likely that not all nodes are part of the network in these larger area configurations. This is further analyzed in Section 6.2.5.

#### 6.2.2. Protocol Overhead

Figure 5 depicts the average energy overhead per node required to construct the broadcast tree in comparison to disseminating data.

On the x-axes, the number of nodes is shown, while the y-axes denote the used average energy per node in Joules. The sub-plots depict the three cycle-handling algorithms, the colors denote different simulation areas, and the line style denotes the energy for broadcast tree construction (dotted) and data dissemination (solid). This plot shows that the energy used is only dependent on the number of nodes and the simulation area, while the energy per node is not significantly influenced by the cycle-handling algorithms. The key takeaway, however, is that disseminating data requires about three times more energy than constructing the broadcast tree in the configuration with many nodes. The fewer nodes that are involved, the greater the energy requirement for disseminating data compared to constructing the broadcast tree. For example, configurations with 50 nodes require up to 17 times more energy for data dissemination. The energy required for disseminating data grows linearly with the number of nodes, whereas the energy required for constructing the broadcast tree grows faster since the coordination and the resulting number of packets do not increase linearly. In summary, the experiments show the feasibility of BTP since the construction of the broadcast tree is only performed once, while the dissemination of data via the ready-to-use broadcast tree can be performed repeatedly afterwards.

#### 6.2.3. Time for Broadcast Tree Construction Phase

Another important metric for the practical use of broadcast trees is the time required to construct a broadcast tree, i.e., the amount of time required until the data can be sent to the nodes. This is shown in Figure 6.

The x-axes show the number of nodes, while the y-axes denote the time taken to construct the broadcast tree in seconds. The simulation area is denoted in the sub-figures, and the cycle-handling algorithms are represented with different colors. The three main properties that significantly influence the time required to construct a broadcast tree are the number of nodes, the simulation area, and the cycle-handling algorithm employed. The more nodes that are in the network and the larger the area, the longer it takes to construct the broadcast tree. Regarding the cycle-handling algorithms, it is noticeable that Ping-to-Source and Path-to-Source show no significant differences. Mutex, however, can take up to 65% longer to construct a broadcast tree. This is because locking an entire sub-tree, switching the parent, and unlocking it again takes time that is not required using the other two algorithms. Due to this behavior exhibited by Mutex, we did not consider it in our real-world implementation.

#### 6.2.4. Cycle Handling

Under certain circumstances, cycles may be either not avoided or detected or are not broken up before the experiment is finished. Figure 7 depicts how many cycles were still present when ending the experiment.

The y-axes show the number of runs in which at least one cycle lasted until the end, while the x-axes show the different numbers of nodes. The sub-plots illustrate the results in the different areas, and the colors denote the three cycle-handling algorithms. It is evident that the number of nodes in the network does not significantly influence the number of cycles lasting until the end of the experiment. Area size, on the other hand, does have a significant influence, especially for the Mutex cycle avoidance algorithm, but also for Path-to-Source. Ping-to-Source is also affected by the area size but significantly less so since cycles could not be detected or broken up in only three out of 1800 Ping-to-Source experimental runs.The Mutex’s behavior can be explained by the problem wherein locking an entire sub-tree leads to an almost complete halt in construction. Suppose the broadcast tree is in a somewhat advanced stage. When one node in the middle of the broadcast tree decides to change its parent, the entire sub-tree will be locked and stops any further optimization until the lock is eventually released. Finally, both cycle avoidance algorithms do not seem to avoid all cycles. This is because they both rely on transmitting information required for the respective algorithm. For Mutex, this transmission entails broadcasting the packet that locks and unlocks sub-trees. When the area is large, it may simply occur that lock or unlock packets are lost or not received by the children, leaving the entire broadcast tree in an unfinished state. The same applies to the Path-to-Source algorithm. This makes Mutex and Path-to-Source effectively unusable in any scenario; therefore, both were not considered in our real-world implementation.

#### 6.2.5. Unconnected Nodes

Figure 8 shows the number of nodes that are not part of the broadcast tree at the end of the experiment.

On the x-axes, the number of nodes is shown, while the three sub-plots depict the three different cycle-handling algorithms. The colors denote different area configurations. The overall ratio of unconnected nodes across all experiments is relatively small, amounting to about 6% in the worst case. However, the cycle-handling algorithms show quite large differences. Ping-to-Source shows good results when only a few nodes are spread out in the area, about 1%, i.e., one node not connected to the broadcast tree (on average). With more nodes in the network, the rate of unconnected nodes decreases to 0.1%. Mutex encounters difficulty when used in large areas, presenting 5% unconnected nodes in area configuration (5) and few nodes. However, while Ping-to-Source and Mutex show reasonable results, Path-to-Source behaves quite erratically. When using the Path-to-Source algorithm, while a node *j* connects to a parent *i*, another node *k* might attempt to connect to *j*, which is the parent of *i*. Since *j* and *i* are not connected yet, the Path-to-Source algorithm fails to detect and avoid this cycle. Therefore, Path-to-Source was not considered in our real-world implementation.

## 7. Real-World Implementation

Since the Matlab and NS-3 simulations in Section 5 and Section 6 provided encouraging results, we also assessed the performance of BTP using real hardware to evaluate its practicability under real-world conditions.

We implemented BTP in userland C for Linux using standard libraries. Thus, we relied on Wi-Fi as the wireless technology since it is widely available on Linux systems. However, as shown in Section 4.1, BTP is not limited to Wi-Fi; it can be used with any radio technology. The SNR calculation of Equation (1) requires information about the wireless transmission characteristics of a received frame. This information, contained in the RadioTap header, is not available to userland C programs or even the Linux kernel since it becomes stripped off by the Wi-Fi chip’s firmware. To bypass this limitation, we used the Nexmon framework [46] (https://nexmon.org (accessed on 4 July 2023)). We created a Nexmon patch that preserves the RadioTap header for MAC frames with our BTP EtherType. All other frames were handled normally to avoid interference with any other programs. Furthermore, we utilized Linux raw sockets for two reasons. First, their use further increases performance since the kernel’s TCP/IP stack is bypassed and a packet is more or less directly passed to the Wi-Fi chip. Second, since BTP is located on the network layer of the ISO/OSI stack, we were able to define a custom EtherType (0x88DF) and enable the reception of BTP packets in userland. Addressing was handled on the Link Layer, with each node identified according to its Wi-Fi MAC address. Neighbour Discovery packets and Application Data packets were sent as MAC-broadcasts; all other packets were unicasts.

In Section 4.2, we discussed the tree construction phase in detail but left the data dissemination phase open to a specific implementation. To reduce the likelihood of collisions, we used an MTU of 1200 bytes, where 37 bytes were used for Ethernet and BTP headers and the remaining 1163 bytes were available for the payload. Furthermore, we added a delay between individual data frames to avoid sending all data frames at once, and the relaying nodes were given a rate limit per sequence number to further reduce the likelihood of collisions and increase the chance of successful data delivery. Additionally, the protocol did not specify what action to take if a node received the entire dataset. In our implementation, leaf nodes do not relay data frames. As soon as a leaf node has received the entire dataset, it disconnects from the parent. This leads to intermediate nodes eventually becoming leaf nodes that also do not relay data frames anymore and disconnect from their parent after receiving the entire data. In this way, the tree is eventually deconstructed.

### 7.1. Testbed

To evaluate BTP, we utilized a testbed deployed at our university. It consists of 75 Raspberry Pis spread over a university building across four floors. Besides Wi-Fi, all nodes also have an Ethernet uplink to a central management server and are configured for network booting to support easy, large-scale deployment. We created a network-bootable image for our experiments using the Pimod framework [47] (https://github.com/Nature40/pimod (accessed on 4 July 2023)). Although we used RAW sockets with a custom EtherType, the Wi-Fi chip did not receive any frames if it was not part of a Basic Service Set (BSS), i.e., associated with an access point or part of the same ad hoc network. Therefore, the Raspberry Pis were set to ad hoc mode with the same BSS ID. Furthermore, the Linux kernel will refuse to send frames to the Wi-Fi interface if an IP address has not been set. Thus, we assigned a random IP address to all Raspberry Pis. We used channel 1 of the 802.11n mode and employed the RTS/CTS method of the 802.11 CSMA/CA mechanism.

### 7.2. Experimental Setup for the Real-World Implementation

Table 4 summarizes the parameters used in our evaluation.

Since we could not alter the position of the nodes (in contrast to our Matlab and NS-3 simulations), we used three source nodes located in the northern part of the building, in its center, and in the southern part. Furthermore, we used three data sizes, representing simple sensor values (1 KiB), network diagnosis (4 KiB), and device updates (16 KiB). Although both simulations showed that 10 iterations constitute a sweet spot between overhead and finding the optimal tree, we sought to evaluate BTP in greater depth for practical implementation. Therefore, we used three finishing thresholds for the counter of the number of iterations without topology changes. Finally, we again compared BTP to SBP. Each experimental configuration was repeated five times, resulting in 270 experimental runs.

### 7.3. Results of the Real-World Implementation

#### 7.3.1. Total Energy Consumption

Due to the testbed’s setup, we could not measure the power draw of all the nodes. Hence, we decided to employ a model to compute the energy consumption in mJ based on the used parameters of the physical and data link layers to estimate the energy used in each run. Figure 9 shows the total energy consumption for different parameter sets. The y-axis shows energy in mJ, while the x-axis denotes different data sizes, where different colors represent different finishing thresholds and SBP, respectively. The values include frames for tree construction as well as the data themselves. BTP requires between 68% (1 KiB and a finishing threshold of 5) and 90% (16 KiB and a finishing threshold of 25) less energy compared to SBP, depending on the data size and the values of the finishing thresholds. This is due to the fact that the broadcast tree resulting from BTP is optimal with respect to the energy requirements. Among the BTP parameters, energy use does not show significant differences with increasing finishing threshold values. While SBP requires about 70% more energy for 16 KiB compared to 1 KiB data, BTP only requires about 20% more energy. This is counter-intuitive since one would expect about the same increase in energy consumption. We discovered that this result was due to RF interference. Neither BTP nor SBP include any advanced MAC mechanism, which leads to congestion in the RF spectrum for larger data sizes. However, because BTP uses less power to transmit data, there is less interference causing retransmission, resulting in a moderate increase in overall energy consumption.

#### 7.3.2. Energy Consumption for Tree Construction and Data Dissemination

Figure 10 presents the energy required for broadcast tree construction and for data dissemination. The x-axis shows different data sizes, while the y-axis shows the energy used for Application Data packets (blue) and for the tree construction packets (red) for BTP and all the finishing thresholds. The energy required for tree construction does not depend on the size of the data but only on the number of nodes; thus, the red plots do not show significant differences. The energy required for data dissemination increases with the size of the data. For the 1 KiB data, about 30% of the energy is used for data dissemination, and about 70% is used for tree construction. However, even in this scenario, BTP performs significantly better than SBP in terms of total energy consumption, as shown in Figure 9. BTP requires about 1.5 MiB of data to construct a tree, regardless of the finishing threshold or data size, whereas data dissemination requires about 150 MiB of total data transfer for 1 KiB experiments, 300 MiB for 4 KiB experiments, and about 400 MiB for 16 KiB experiments. Note that there are outliers in experiments, since the environment can vary under real conditions. For example, in the 4 KiB experiments, three runs required between 740 mJ and 800 mJ of energy, which is outside the 1.5-fold interquartile range.

#### 7.3.3. Successful Receptions

Figure 11 shows the results in terms of successful deliveries. The x-axis shows experiments with different data sizes, while the y-axis shows the percentage of nodes that received the entire dataset. The colors denote different finishing thresholds and SBP, respectively. BTP produces results that are at least on par with SBP; with larger data, SBP’s performance becomes even worse. For 1 KiB and 4 KiB data sizes, BTP achieves a nearly 100% delivery ratio, with a few runs producing outliers. SBP only achieves this success ratio for 1 KiB, while for 4 KiB and 16 KiB data sizes, the average delivery rate falls below 85%. For a data size of 16 KiB, BTP still achieves an average delivery ratio of about 98%. This result (and the failed runs for 1 KiB and 4 KiB data sizes) is mainly due to the varying conditions of the network and the wireless medium. Even though the tests were conducted in a testbed, there were no lab conditions. The testbed was deployed in a building in the university, where there are offices and lecture rooms with employees and students and a number of other Wi-Fi networks; thus, some of the experiments were not completed. The poor performance of SBP, however, is due to the fact that BTP makes better use of the wireless medium by refraining from flooding the network with maximum transmission power, thus producing less interference between stations. Note that there are outliers in experiments, since the environment can vary under real conditions. For example, in the 1 KiB and 16 KiB experiments, there are runs that have no successful receptions at all, which is outside of the 1.5-fold interquartile range.

#### 7.3.4. Time for Broadcast Tree Construction Phase

Figure 12 shows the time taken to construct the broadcast tree. The x-axis shows the different data sizes, while the y-axis shows the construction time in seconds. The different colors represent different finishing thresholds. This figure has three key takeaways. First, the size of the data does not influence the time it takes to construct the broadcast tree. Second, the higher the finishing threshold, the longer it takes to construct the tree, which is the expected result. Third, the average time of seven seconds for the initial construction of the broadcast tree, which is only performed once, is reasonable since the constructed broadcast tree can subsequently be used for an arbitrarily long time period. Note that there are outliers in experiments, since the environment can vary under real conditions. For example, in the 1 KiB SBP experiments, there are runs that take up to 40 s to construct the tree, which is outside of the 1.5-fold interquartile range.

#### 7.3.5. Contributions of Individual Nodes

Our last evaluation shows how much energy is contributed by individual nodes in the network. Figure 13 is a scatter plot where every dot represents the proportional energy used by a node in a particular experimental run.

The x-axis arranges nodes according to their loads (not their topological relationships) in relation to the total energy consumed over the entire experiment. This means that, for example, node 1 is not next to node 2, but node 1 contributes more to the total amount of energy required than node 2. The y-axis shows how much an individual node contributes to the amount of energy required, e.g., a dot at 50% at 0 means that node 0 contributes 50% of the required energy. The sub-plots and colors represent the two protocols with all experimental configurations. Using BTP, only a few nodes had a relatively high level of energy usage, while the nodes beyond node 20 barely contributed to the overall level of energy consumption. In SBP, on the other hand, all nodes contributed to overall energy utilization. This shows that although there are nodes in BTP that have a high load, the energy for most nodes is preserved, giving them a higher lifetime when battery-powered. Note that Figure 13 is a scatter plot covering all experimental runs; thus, it is not easy to see which values sum to 100%. There were experiments where, taking SBP as an example, a single node was responsible for 100% of the energy consumed. In this specific experiment, no other node sent any data, indicating that one node in the neighborhood of the source node received the data. This was the case for the unfinished experiments with a 1 KiB data size and a finishing threshold of 5, as indicated in Figure 11.

## 8. Conclusions

In this study, we presented BTP, a novel broadcasting protocol for wireless multi-hop networks based on a game-theoretical model and designed to function in practical implementations. BTP constructs a spanning tree in a decentralized manner to minimize the total energy consumption of an entire network by minimizing the transmission power at each node. In this section, we highlight our contributions as well as directions for future work.

### 8.1. Contributions

We made the following contributions. First, we adopted a game-theoretical model for the design of BTP while preserving the relevant convergence characteristics of the game-theoretical model. Second, we addressed the practical constraints in the implementation of BTP, i.e., (a) each node operates only on information that is locally available in its neighborhood, (b) all nodes operate asynchronously without requiring any form of time synchronization, and (c) ping-pong effects possibly occurring in the original game-theoretical model are avoided. Third, we integrated three algorithms capable of inhibiting the creation of graph cycles into the design of BTP. Finally, we evaluated BTP with respect to various aspects. We performed simulations to compare BTP to other algorithms from the literature and investigate the scalability of BTP. A practical implementation of BTP on a testbed with 75 Raspberry Pis allowed us to evaluate BTP under realistic conditions. BTP was able to achieve a total energy reduction of up to 90% compared to a simple broadcast protocol in our testbed.

### 8.2. Future Work

There are several areas for future work. For example, the current BTP implementation does not consider node mobility, which would require the spanning tree to be maintained continuously, including during the data dissemination phase. Furthermore, BTP can be enhanced with mechanisms that increase the reliability of data transfers, such as acknowledgments or checksums. In its current form, BTP also only allows data dissemination from the source node to the other nodes. However, as soon as a broadcast tree is constructed, bi-directional data transfer should be supported, e.g., to allow the source node to function as a data sink for sensor nodes. Moreover, once the broadcast tree has been built, routing protocols such as AODV [48] or DSR [49] may be used to transmit data to a particular node. Finally, BTP is based on a conventional medium access control mechanism with random back-off times, which works well in many scenarios. Future work might also incorporate coordinated medium access control mechanisms to provide real-time guarantees and further enhance the reliability of BTP [50].

## Figures and Tables

**Figure 1 sensors-23-07419-f001:**
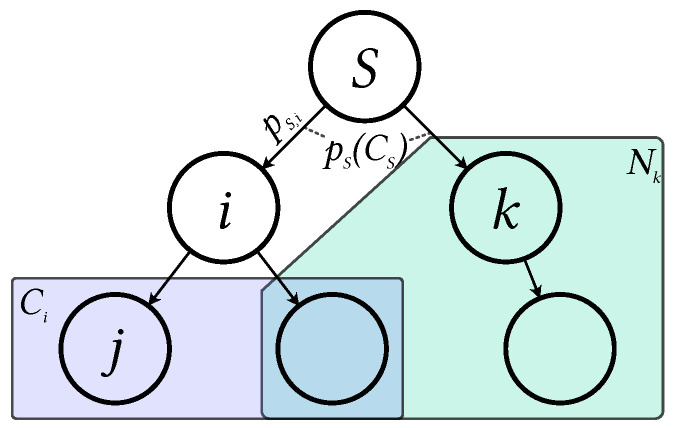
Broadcast tree overview.

**Figure 2 sensors-23-07419-f002:**
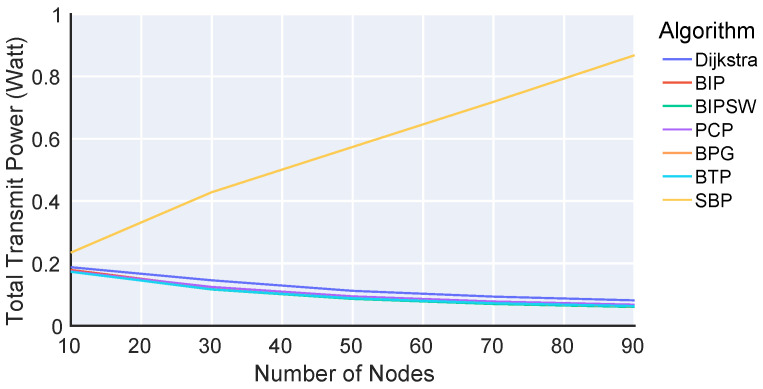
Total required transmission power for various algorithms.

**Figure 3 sensors-23-07419-f003:**
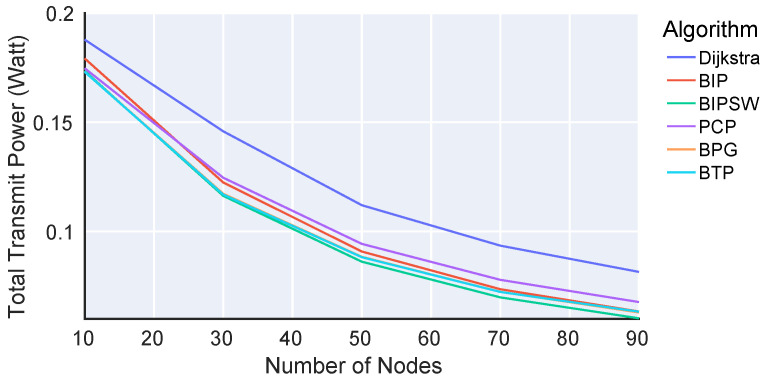
Total required transmission power for various algorithms (without SBP).

**Figure 4 sensors-23-07419-f004:**
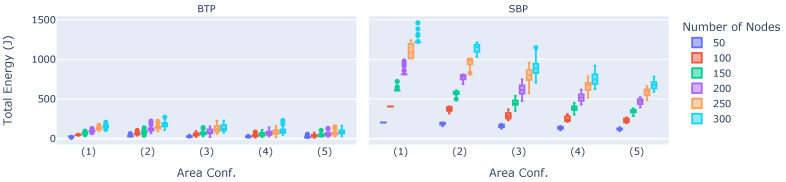
Total energy consumption values of BTP and SBP.

**Figure 5 sensors-23-07419-f005:**
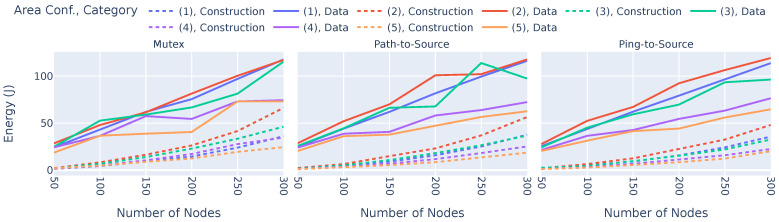
Energy usage of broadcast tree construction phase and data dissemination phase.

**Figure 6 sensors-23-07419-f006:**
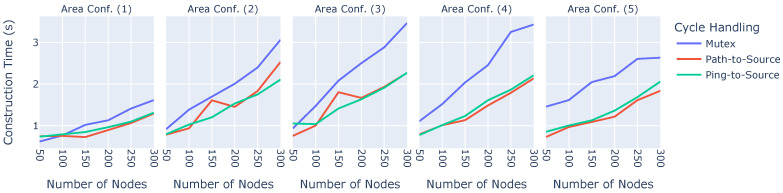
Time taken to construct the broadcast tree.

**Figure 7 sensors-23-07419-f007:**
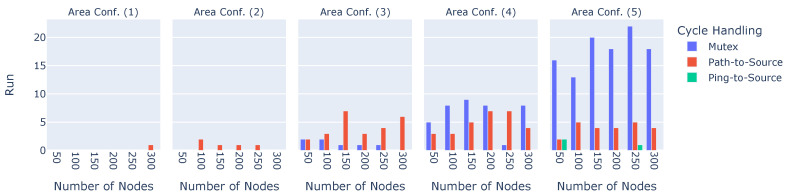
Number of cycles lasting until the end of an experiment.

**Figure 8 sensors-23-07419-f008:**
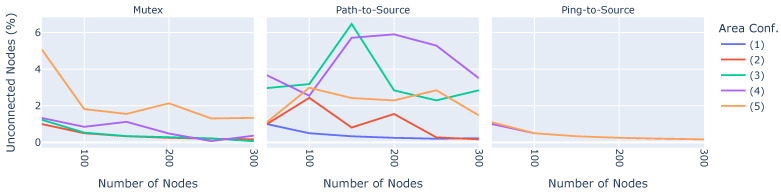
Percentage of nodes not part of the broadcast tree.

**Figure 9 sensors-23-07419-f009:**
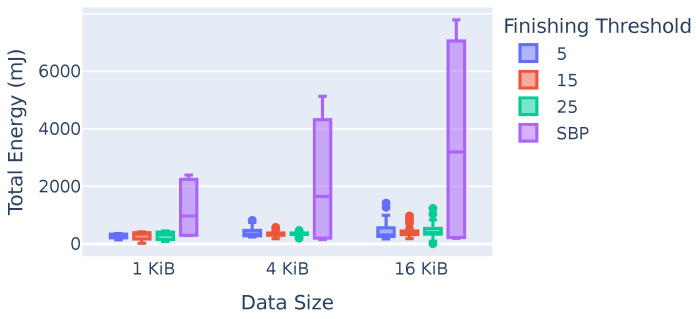
Total required energy.

**Figure 10 sensors-23-07419-f010:**
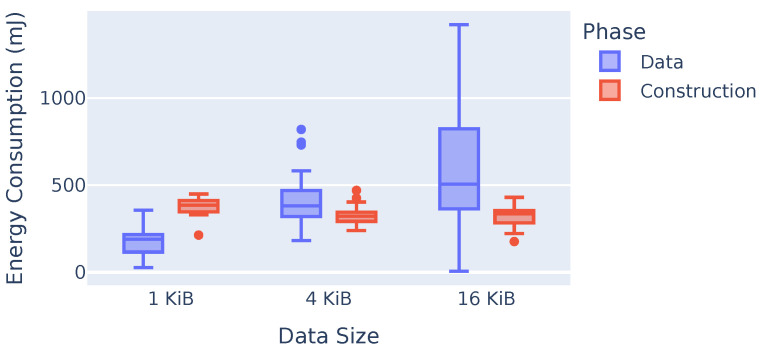
Energy required for both broadcast tree phases.

**Figure 11 sensors-23-07419-f011:**
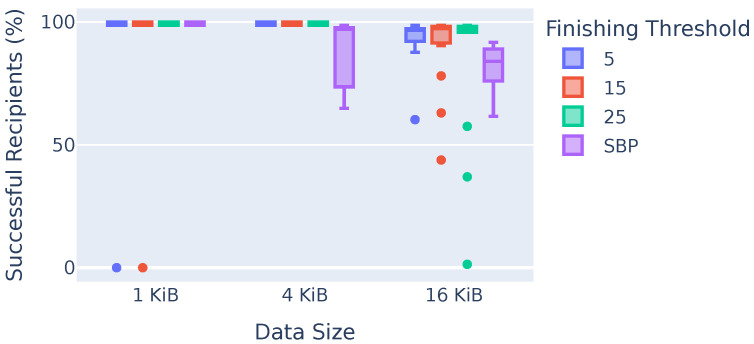
Ratio of nodes successfully receiving data.

**Figure 12 sensors-23-07419-f012:**
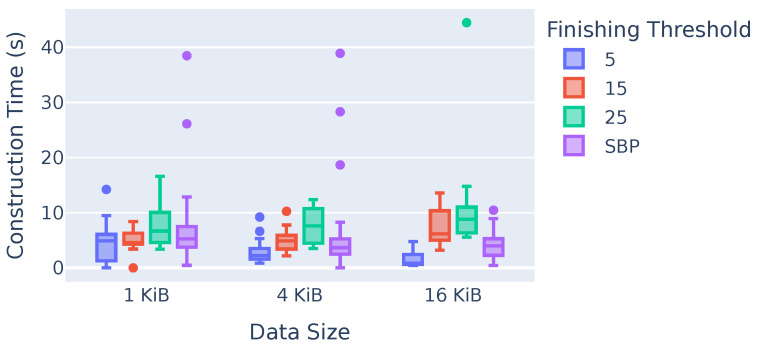
Time taken to construct the broadcast tree.

**Figure 13 sensors-23-07419-f013:**
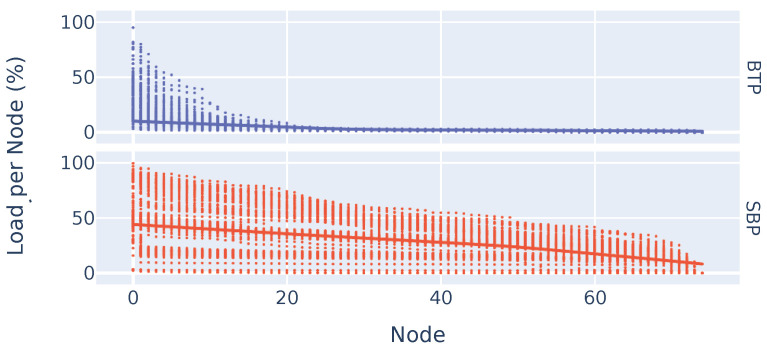
Scatter plot of the load distribution per node over different experimental runs.

**Table 1 sensors-23-07419-t001:** Mathematical notations.

Notation	Description	Notation	Description
*T*	Broadcast tree consisting of nodes *V* and edges *E*	v∈V	A node of the broadcast tree *T*
S∈V	Source node of the broadcast tree	e∈E	An edge of the broadcast tree *T*
$p^{\max}$	Maximum possible Tx power	$p_{i}$	Tx power of node i∈V
$|h_{i,j}{|}^{2}$	Channel gain of (i,j)∈E	$\sigma^{2}$	Noise power
$\gamma_{j}$	SNR at node *j*	$\gamma^{min}$	Minimum required SNR
$C_{i}$	All children of node i∈V	$N_{k}$	All neighbours of node k∈V
$p_{i,j}$	Tx power that node i∈V must use to reach node j∈V	$p_{i}\left(C_{i}\right)$	Tx power that node i∈V must use to reach all its children $C_{i}$
*p*	Sum of Tx powers of all nodes in *V*		
G	Potential game	P	Set of rational players V∖{S}
$a_{j}^{\left(t\right)}$	Selected parent of destination node *j* at iteration *t*	$a^{\left(t\right)}$	Action profile at iteration *t*
$\varphi_{j}\left(a_{j}^{\left(t\right)}\right)$	Cost function of destination node *j* for action $a_{j}^{\left(t\right)}$ at iteration *t*	$A_{j}^{\left(t\right)}$	Set of possible actions (parents) for node *j* at iteration *t*

**Table 2 sensors-23-07419-t002:** Parameters used for the Matlab simulation.

Parameter	Values
Protocols	BTP, BPG, BIP, BIPSW, PCP, SBP, Dijkstra
Nodes	10, 20, 30, 40, 50, 60, 70, 80, 90
Simulation Area	500 m × 500 m
$p^{\max}$	20 dBm
$|h_{i,j}{|}^{2}$	$\frac{1}{d^{\alpha }}$
α	3
$\gamma^{min}$	10
$\sigma^{2}$	−90 dBm
Finishing Threshold	10

**Table 3 sensors-23-07419-t003:** Parameters used for the NS-3 simulation’s evaluation.

Parameter	Values
Protocols	BTP, SBP
Cycle Handling	Ping-to-Source, Path-to-Source, Mutex
Nodes	50, 100, 150, 200, 250, 300
Simulation Area	(1) 100 m × 100 m, (2) 200 m × 200 m, (3) 300 m × 300 m, (4) 400 m × 400 m, (5) 500 m × 500 m
Data Size	1 KiB
Finishing Threshold	10
Runs	30

**Table 4 sensors-23-07419-t004:** Parameters used in our evaluation.

Parameter	Values
Source Nodes	3
Data Sizes	1 KiB, 4 KiB, 16 KiB
Finishing Threshold	5, 15, 25
Protocols	BTP, SBP
Runs	5

## Data Availability

Publicly available datasets were analyzed in this study. This data can be found here: https://uni-marburg.de/nXNAEL (accessed on 4 July 2023). Furthermore, the code written for the protocol, the NS-3 simulation, and the experiments can be found in the following Github repositories: https://github.com/umr-ds/broadcast-tree (accessed on 4 July 2023), https://github.com/umr-ds/broadcast-tree-protocol (accessed on 4 July 2023), https://github.com/umr-ds/broadcast-tree-ns3 (accessed on 4 July 2023).

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
