# Peer review of "Energy-Efficient Decentralized Broadcasting in Wireless Multi-Hop Networks [Author-notes fn1-sensors-23-07419]"

_sensors, 2023, doi:10.3390/s23177419_

Round 1

Reviewer 1 Report

1.Why is the applicability of game theory model limited in practice?

2.What is the weak advantage strategy and how does it help to avoid the ping-pong effect?

3.What are the types of BTP cycle avoidance and detection algorithms? Please further introduce and evaluate them.

4. The implementation of reality, when the child node can reach two potential parent nodes with the same transmission, the power and will switch repeatedly between them, why can't you find a problem that may appear stable broadcast tree T?

5. How does BTP achieve energy minimization and proliferation?

6. What is the mechanism of reliable data transmission mentioned in future work?

7. In the Introduction, when summarizing the work of predecessors, we should pay attention to the shortcomings of the work completed by our predecessors. In the Conclusion, we should emphasize the advantages and innovations of our own work compared with those of predecessors.

Author Response

Response to Reviewer 1

We would like to express our gratitude to the reviewers for the time invested in thoroughly reading our paper and for their valuable suggestions to improve it. We appreciate the questions raised. We have done our best to address all the reviewers' comments and have accordingly revised the manuscript as suggested. The sections and text passages that were changed in accordance with Reviewer 1's comments are shown in blue in the revised manuscript. Detailed responses to each comment can be found below.

Detailed Responses

1. Why is the applicability of game theory model limited in practice?

We thank the reviewer for this question. In the introduction of the original manuscript, we only briefly mentioned the shortcomings and impractical assumptions of the original model and assumed that the detailed discussion in Section 4.1.2 was sufficient. In the Introduction on page 2 of the revised manuscript, we discuss the shortcomings of the original model in more detail and explicitly refer to Section 4.1.2 to elaborate how we address these shortcomings.

2. What is the weak advantage strategy and how does it help to avoid the ping-pong effect?

We thank the reviewer for the suggestion to explain the ping-pong effect in more detail and how to avoid it. First of all, however, it must be clarified that the introduction of the original manuscript referred to a weakly dominant strategy.
We made a mistake here. The original model of Mousavi et al. on which BTP is based uses a weakly dominant strategy, which can result in ping-pong effects.
In the course of our approximation, we changed this strategy to a strictly dominant strategy. This ensures that a node only connects to another node when the global transmission power becomes strictly smaller, rather than remaining smaller or the same as in a weakly dominant strategy. The explanation of the ping-pong effect can be found in the response of comment 1 in the Introduction on page 2 of the revised manuscript.

3. What are the types of BTP cycle avoidance and detection algorithms? Please further introduce and evaluate them.

Thank you for this comment. These algorithms are discussed in detail in Chapter 4.2 and evaluated in Chapter 6.2. Nevertheless, in the revised manuscript, we summarize the three algorithms on page 2 of the Introduction (4th contribution) to provide a better understanding.

4. The implementation of reality, when the child node can reach two potential parent nodes with the same transmission, the power and will switch repeatedly between them, why can't you find a problem that may appear stable broadcast tree T?

Thank you for the question, but we do not fully understand the problem.
The original model of Mousavi et al. uses a weakly dominant strategy, as explained in the Introduction and in Section 4.1. That is, given two potential parent nodes that do not change the total transmission power no matter which one a child connects to, that child node may constantly switch between these two potential parent nodes and never converge to a stable solution. This condition is referred to as the ping-pong effect. Therefore, in our implementation, we changed the strategy to a strictly dominant strategy that prevents this problem. With our approximation, if there were two potential parent nodes that had the same effect on the total transmission power, the child node would no longer switch, thus avoiding the ping-pong effect. These changes and their effects are discussed in Sections 4.1.1 and 4.1.2, and in the revised manuscript also in the sections highlighted with blue color in the Introduction.

5. How does BTP achieve energy minimization and proliferation?

Thank you for highlighting the omission regarding the elaboration of the energy minimization mechanism. While the Abstract and Section 4 explain the main mechanism, we neglected to mention it in the Introduction. The Introduction of the revised manuscript now also includes this aspect on page 2. The gist is to build the tree in such a way that each node adjusts its own transmission power to minimize the contribution to the global transmission power.

6. What is the mechanism of reliable data transmission mentioned in future work?

Thank you for pointing out this shortcoming. We refer to mechanisms typically found on the transport layer of the ISO/OSI model, such as acknowledgments or checksums, mechanisms that allow reliable delivery of data even if underlying layers are lossy or error-prone. In addition, we highlight coordinated medium access as a means to enhance the reliability of BTP in the Future Work section of the Conclusion.

7. In the Introduction, when summarizing the work of predecessors, we should pay attention to the shortcomings of the work completed by our predecessors. In the Conclusion, we should emphasize the advantages and innovations of our own work compared with those of predecessors.

Thank you for raising this issue. We now give an overview of the shortcomings of approaches from the literature in the Introduction of the revised manuscript, starting on page 1. Furthermore, the Conclusion on page 20 now also summarizes the contributions of this paper.

Reviewer 2 Report

Minor editing of English language is needed.

Author Response

Response to Reviewer 2

We would like to thank the reviewer for taking the time to thoroughly read our paper and for their valuable suggestions on how to improve it. We appreciate the issues raised. We have done our best to address all of the reviewers' comments and have revised the manuscript accordingly. The sections and text passages that have been changed as a result of Reviewer 2's comments are shown in red in the revised manuscript. Finally, we treat comments 4 and 6 of this reviewer together because their content belongs together. Detailed responses to each comment are provided below.

Detailed Responses

1. In abstract, it is suggested to briefly introduce the background or facing challenges at the beginning, as the main work in this current version only occupy half.

Thank you for highlighting the lack of the addressed challenges in the Abstract.
We rephrased a part of the Abstract, as indicated by the red color on page 1. Since we cannot explicitly describe the challenges in detail in the Abstract for the sake of brevity, the revised text indicates these challenges implicitly. In particular, we indicate that BTP does not depend on certain idealized assumptions made by the game-theoretical model (i.e., global knowledge about other players' actions, global time synchronization of all nodes) and that BTP is still able to inhibit graph cycles (that would occur if not properly addressed).

2. The authors claimed that “One way of achieving minimal energy consumption across a network in a multi-hop manner is to utilize a spanning tree topology with the minimally required transmission power at each node.” What is the difference between the proposed BTP and the existing spanning tree topology? The advantages also need to be emphasized.

Thank you for pointing out that there is a problem with this sentence. In fact, the wording of this sentence was probably misleading as there is no existing spanning tree topology upon which we rely. Therefore, we have changed the corresponding text passage in the Introduction on page 1 of the revised manuscript such that it first motivates the need for a spanning tree to minimize the energy consumption. Then, we emphasize the associated challenges in constructing an energy-minimal spanning tree in practice.

3. The authors should use the past tence to introduce the Related Work section.

Thank you for pointing out this issue. We have carefully read the Related Work chapter and adapted the relevant passages. However, these changes are not marked separately in red, since the changes were made selectively throughout the chapter.

4. It is suggested to introduce the following advanced techniques in energy efficiency optimization [R1]-[R2], wireless multi-hop networks [R3] and IoT [R4] fields to highlight the state-of-art of this paper:

Thank you for the proposed works. We have added a paragraph in the Related work regarding mechanisms other than topology control to increase energy efficiency, such as satellite communications and beamforming. However, we think that these topics, although important to their respective domains, are not central to multi-hop wireless sensor networks.

6. It is also suggested to introduce some techniques, such as beamforming, in related works, which is an efficient way to minimize the power consumption.

Thank you for this useful hint. We have added a paragraph at the end of the Related Work section that discusses mechanisms other than topology control related to energy efficiency. However, we do not think that these aspects, although important and relevant in their respective fields, are central to multi-hop wireless sensor networks. This is also stated in the corresponding red text in the Related Work section.

5. Both the motivations and contributions need to be further clarified to demonstrate that why the authors investigated this work and what is the novel points in this paper compared to other works.

Thank you for this suggestion. Please note that this issue was also pointed out by other reviewers. Therefore, in the revised manuscript, we highlight the contributions in the Abstract and in the Conclusion with red and blue text colors, respectively. In addition, the Introduction now lists the contributions more explicitly as a list of items. The respective text passages were also revised to better highlight the specific contributions of each point. This change can be seen in the Introduction on page 2 of the revised manuscript.

7. The full name of the benchmark schemes should be provided and detailed introduced.

We thank the reviewer for pointing out this shortcoming. Indeed, we missed to describe these algorithms and approaches from the literature. We now provide more details about each algorithm in the red text on page 11 of the revised manuscript.

Reviewer 3 Report

This paper is well written and contributions are good enough for this journal. The extensions over the conference paper are acceptable. However, the following minor changes maybe done before consider this paper for publication.

1. There are several notations throughout the paper, which are summarized using a table for easy to refer.

2. The authors may consider existing datasets for experiments such as -- EDGF: Empirical dataset generation framework for wireless sensor networks

3. Provide the mathematical formulae for the metrics evaluated in this work.

4. What are the primary reasons noticed by the authors to achieve superior performance over the existing ones?

5. How does this work can be extended in the near future?

6. Which applications are more appropriate to use this work in realtime scenarios?

7. What are the major challenges pose when implement this work in realtime scenarios?

Author Response

Response to Reviewer 3

We thank the reviewer for the careful review of our manuscript and for their insightful recommendations to improve its quality. The concerns raised have been carefully taken into account. Every effort has been made to incorporate the reviewers' suggestions, resulting in substantial revisions throughout the manuscript. For the ease of understanding, the revised version highlights in green the parts that have been changed in response to reviewer 3's comments. Comprehensive responses to each comment are presented below.

Detailed Responses

1. There are several notations throughout the paper, which are summarized using a table for easy to refer.

Thank you for that suggestion. We added the table with mathematical notations at the start of Section 3 on page 4 of the revised manuscript.

### 2. The authors may consider existing datasets for experiments such as -- EDGF: Empirical dataset generation framework for wireless sensor networks

Thank you for that insightful suggestion. To generate our sample data for evaluation, we made the following considerations. First, about node placement.
In the two simulations, we intentionally used uniformly distributed random placement to represent as many different scenarios as possible given the large number of simulation runs (63,000 for Matlab and 5,400 for NS-3). This results in a particularly broad picture in the results that is not dependent on any particular node placement. We have added this argument as an explanation on pages 10 and 13 of the revised manuscript. For the real-world evaluation, we could not randomly place the nodes because we relied on the local conditions of the university building. Second, regarding the disseminated data, the focus of both simulative evaluations was on the construction of the tree. The data dissemination phase was only used to validate that the constructed tree leads to an optimal result comparable to the literature. For the evaluation of the real implementation, however, we did not choose random data sizes, but deliberately focused on the three chosen (1 KiB, 4 KiB, 16 KiB), since these are data sizes commonly found in WSNs.

### 3. Provide the mathematical formulae for the metrics evaluated in this work.

Thank you for this suggestion. We added the formulae to Figures 2 and 3 since these figures show the main motivation for this paper, i.e., minimizing the total required energy consumption. However, we think that adding formulae to the remaining figures would add confusion and dilute the reasoning, since we would need to introduce new mathematical symbols and corresponding explanations.

### 4. What are the primary reasons noticed by the authors to achieve superior performance over the existing ones?

Thank you for pointing out this shortcoming. In the discussion of Figures 2 and 3 on pages 11 and 12, we now give a short explanation of the respective results, so that it is a discussion and not only a description of the figures.

5. How does this work can be extended in the near future?

Thank you for that question. Although there was already a future work paragraph in the original version of the manuscript, we added a headline in the Conclusion section of the revised manuscript, so that this point will not be overlooked and can be easily found.

6. Which applications are more appropriate to use this work in realtime scenarios?

We thank the reviewer for this question. However, it is not entirely clear to us what the reviewer means by realtime. BTP is designed to distribute data such as network configuration, updates, or notifications live and without further processing. This makes BTP a realtime protocol in a broader sense.
However, BTP does not provide realtime guarantees in the sense that data reaches its destination within a strict deadline. These aspects are beyond the scope of our work and will not be discussed further.

7. What are the major challenges pose when implement this work in realtime scenarios?

We thank the reviewer for this question. As already discussed in the response of reviewer 3's question 6, realtime systems are out of scope of this work. However, we added this aspect to the future work section of the conclusion in the revised manuscript.

Round 2

Reviewer 2 Report

The authors have addressed all my concerns, no further comments.

Author Response

Thank you for your feedback and your dedication to improving our manuscript and research in general.